# Optimization of Protoplast Preparation Conditions in *Lyophyllum decastes* and Transcriptomic Analysis Throughout the Process

**DOI:** 10.3390/jof10120886

**Published:** 2024-12-21

**Authors:** Xiaobin Li, Ying Qin, Yufei Kong, Samantha Chandranath Karunarathna, Yunjiang Liang, Jize Xu

**Affiliations:** 1College of Agriculture, Yanbian University, Yanji 133002, China; xiaobinlee@163.com (X.L.); kong4523@126.com (Y.K.); 2College of Agriculture, Jilin Agricultural Science and Technology University, Jilin 132000, China; 19975669975@163.com; 3College of Forestry, Beihua University, Jilin 132000, China; 4Center for Yunnan Plateau Biological Resources Protection and Utilization, Qujing Normal University, Qujing 655011, China; samantha@mail.kib.ac.cn

**Keywords:** functional genomics, fungi, protoplast regeneration, RNA-seq

## Abstract

Protoplasts are essential tools for genetic manipulation and functional genomics research in fungi. This study systematically optimized protoplast preparation conditions and examined transcriptional changes throughout the preparation and regeneration processes to elucidate the molecular mechanisms underlying the formation and regeneration of protoplasts in *Lyophyllum decastes*. The results indicated an optimal protoplast yield of 5.475 × 10^6^ cells/mL under conditions of fungal age at 10 days, digestion time of 2.25 h, enzyme concentration of 2%, and digestion temperature of 28 °C. The Z5 medium supplemented with *L. decastes* mycelial extract achieved a high regeneration rate of 2.86. RNA-seq analysis revealed 2432 differentially expressed genes (DEGs) during protoplast formation and 5825 DEGs during regeneration. Casein kinase I, cytochrome P450 (CYP52), and redox-regulated input receptor (PEX5) were significantly upregulated during the protoplast stage, while β-1,3-glucan synthase (SKN1), chitin synthase (CHS2), hydrophobin-1, and hydrophobin-2 showed significant upregulation during the protoplast regeneration phase. These findings provide a reference for the efficient preparation and regeneration of protoplasts and offer new insights into the molecular mechanisms of protoplast formation and cell wall regeneration in fungi.

## 1. Introduction

*Lyophyllum decastes*, commonly referred to as the fried chicken mushroom, is a rare edible fungus renowned for its exceptional nutritional value. Belonging to Basidiomycota, Agaricales, Lyophyllaceae [1], it is characterized by its thick and delicate flesh, distinctive aroma, and excellent taste. This mushroom is rich in crude protein, amino acids, B vitamins, and trace elements like zinc, copper, and selenium [2,3], making it a valuable addition to any diet. Its health-promoting properties, including anti-tumor activities [4], hypoglycemic effects [5], and anti-hypertensive properties [6], hint at its significant market potential. China’s successful year-round facility cultivation of this species in 2006 and the establishment of mechanized and industrialized cultivation in 2013 [7] have paved the way for its production growth. The production of *L. decastes* in China reached 960 tons in 2014 and experienced rapid growth after 2015 [8,9]. According to the data of the public service platform of the China Edible Fungi Association (https://bigdata.cefa.org.cn/output.html, accessed on 12 December 2024), the production reached 107,601 tons in 2022. Recent discoveries of its immunomodulatory [10], hepatoprotective [11], and weight-reducing effects [12] have only added to its nutritional appeal.

Protoplasts, cells devoid of cell walls while maintaining basic cellular functions and totipotency, are essential for fungal genetic manipulation. They hold the potential for exciting breakthroughs in cell fusion and functional genomic studies [13]. In the realm of edible and medicinal fungi, protoplast technology is a promising avenue for strain improvement and genetic engineering [14]. The efficient preparation and regeneration of fungal protoplasts are crucial prerequisites for their practical applications. Previous studies have demonstrated that multiple factors influence protoplast yield and viability, including enzyme concentration, osmotic stabilizers, enzymatic temperature, digestion duration, and mycelial age. However, the lack of cell wall protection makes protoplasts highly susceptible to lysis, directly affecting their regeneration rates. Notably, research on protoplast preparation and regeneration in *L. decastes* is a task of urgent importance that still needs to be completed. Recent advances in genomics [15] and functional genomics (including transcriptomics and proteomics) have significantly enhanced our understanding of cell wall regeneration mechanisms. In plant systems, dynamic transcriptional changes during cell wall removal and regeneration have been revealed in rice suspension cells [16], identifying several essential regulatory genes, including four glycoside hydrolases (α-amylase, β-amylase, chitinase, and GH36 family genes), three kinases, three zinc finger proteins, phenylalanine ammonia-lyase, and phosphodiesterase. The upregulation of multiple cell wall-related proteins was demonstrated during cotton protoplast cultivation, such as proline-rich protein (PRPL), glycine-rich protein (GRP), extension proline-rich 1 (EPR1), fasciclin-like arabinogalactan protein 2 (FLA2), and expansin-like proteins A and B (EXLA/B) [17]. In fungi, current research has primarily focused on optimizing preparation conditions, including enzyme combinations, osmotic stabilizers, enzymatic temperature, digestion duration, and mycelial age [18,19,20]. However, molecular mechanisms governing protoplast formation and cell wall regeneration remain largely unexplored. This study optimized protoplast preparation conditions for *L. decastes*, screened suitable regeneration media, and analyzed transcriptome profiles during protoplast preparation and regeneration using RNA-seq technology. These findings will provide new insights into improving protoplast preparation techniques and understanding fungal cell wall biology in *L. decastes*.

## 2. Materials and Methods

### 2.1. Bacterial Strains and Culture Conditions

The fruiting bodies of *L. decastes* utilized in this research were sourced from the Qipan Mountain National Scenic Tourist Area, located in Shenyang, Liaoning Province. The strain “HMJU 6921” was isolated from tissue samples, identified, and stored at the Fungal Resource Development and Utilization Laboratory of Jilin Agricultural Science and Technology College. It was cultivated on a potato dextrose–agar (PDA) medium and maintained at a stable temperature of 25 °C under dark conditions for 10 days. After incubation, small sections of the cultured strain, approximately 6 mm × 6 mm in size, were excised using a sterile scalpel, transferred into 100 mL of potato dextrose broth, and incubated in darkness at 25 °C while shaken at 120 rpm.

### 2.2. Protoplast Preparation

Under sterile conditions, the filtered mycelia (using sterile double gauze to filter) were washed twice with sterile water and an osmotic stabilizer. (The osmotic stabilizer was prepared by dissolving the corresponding inorganic salt in sterile water at a concentration of 0.6 mol·L^−1^, sterilized at 121 °C and 0.1 MPa for 15 min and then used: Aladdin™ S112236 Sucrose 99.5%, Aladdin™ D657382 Mannitol 97%, Aladdin™ C111538 Sodium chloride 99.5%, Aladdin™ P301833 Potassium chloride 99%, and Aladdin™ M119381 Magnesium sulfate anhydrous 99%). The mycelia were then placed on filter paper to absorb excess moisture, and approximately 200–300 mg was weighed and transferred into a 2 mL centrifuge tube. Next, 1.5 mL of enzyme solution (lywallzyme, Guangdong Institute of Microbiology, Guangzhou, China) was added to the sample, which was then gently mixed for incubation at a constant temperature. During the incubation, the tube was inverted every 30 min to facilitate protoplast release. After enzymatic digestion, the crude protoplast extract was obtained. The top of a 2 mL glass syringe was plugged with approximately 1 cm of sterile absorbent cotton. After sterilization, it was used to filter the extract and centrifuged at 2500 r·min^−1^ for 10 min, and the supernatant was discarded. The pellet was washed twice with 0.6 mol·L^−1^ osmotic stabilizer to remove any remaining enzyme solution. Finally, 1 mL of 0.6 mol·L^−1^ osmotic stabilizer was added to resuspend the pellet, resulting in the protoplast stock solution. The protoplasts were then diluted to the appropriate concentration and counted using a hemocytometer with a 400× microscope (BX53F, Olympus, Tokyo, Japan).

### 2.3. Single-Factor Experiment

Following the aforementioned method, the effects of different enzymatic digestion temperatures, digestion times, fungal ages, osmotic stabilizers, and enzyme solution concentrations (Mass concentration *m*/*v*) (Table 1) on protoplast yield were investigated. Each experiment was repeated at least three times, using the protoplast yield as the indicator.

### 2.4. Orthogonal Experiment

Based on the results of the single-factor experiments, orthogonal experiments with four factors and three levels were designed using SPSS 26.0 to optimize protoplast preparation conditions. The factors include fungal age, digestion time, digestion temperature, and enzyme solution concentration, with protoplast yield as the evaluation criterion (Table 2).

### 2.5. Screening of Protoplast Regeneration Medium

The protoplast stock solution was adjusted to a suitable concentration by diluting it with a 0.6 mol·L^−1^ osmotic stabilizer. Subsequently, 100 μL of this diluted solution was evenly distributed across various regeneration media (refer to Table 3), while the PDA medium served as the control. Both the experimental and control setups were maintained at 25 °C in darkness. Daily observations were made to monitor colony regeneration, and the regeneration rate was determined. Each experiment was performed a minimum of three times, and the regeneration rate was calculated using the following formula:Regeneration rate = (Number of colonies on regeneration medium − Number of colonies on PDA medium)/Number of protoplasts plated × 100%.(1)

### 2.6. Transcriptome Sample Preparation

The samples used in this study were collected from three different stages of *L. decastes* protoplast preparation: undecided mycelium stage (LDY1), protoplast stage (LDY2), and protoplast-regenerated mycelium stage (LDY3) (Figure 1). Samples were randomly collected three times from each stage, resulting in a total of nine samples. The mother strain was inoculated onto the PDA medium and incubated at 25 °C for 10 days. After incubation, the strain was cut into small pieces (approximately 6 mm × 6 mm) using a sterile scalpel and inoculated into the potato dextrose liquid medium. The culture was incubated at 120 rpm and 25 °C in the dark for 10 days, after which the mycelia were filtered and collected (LDY1). Protoplasts were prepared using mycelia cultured for the same number of days, following the previously described method. After enzymatic digestion, the protoplasts were obtained through filtration using sterile absorbent cotton, followed by centrifugation to remove excess digestion solution. Trizol was immediately added, and the mixture was shaken thoroughly (LDY2). The prepared protoplasts were then diluted to an appropriate concentration and spread evenly onto the regeneration medium. Approximately 15 days later, the regenerated protoplasts started to produce mycelia, which were then scraped off for collection (LDY3). All samples were immediately flash-frozen in liquid nitrogen and stored at −80 °C.

### 2.7. Transcriptome Sequencing, Assembly, DEG Functional Annotation and qRT-PCR Validation

All experimental protocols, including RNA extraction, quality assessment, library preparation, de novo transcriptome assembly, DEG analysis, functional enrichment and qRT-PCR verification, were performed according to the original protocol of Tiantian Wang [23] without modification. The complete protocol is included in the Appendix A, and the qRT-PCR primer sequences were placed in Appendix A. References [24,25] are cited in the Appendix A.

### 2.8. Statistical Analysis

An orthogonal experimental design and a range analysis of the results were performed using IBM SPSS Statistics 26. A single-factor analysis of variance (ANOVA) was conducted with GraphPad Prism 10, which was also utilized for figure generation.

## 3. Result

### 3.1. Single-Factor Experiment for Protoplast Preparation of L. decastes Strain HMJU 6921

#### 3.1.1. Effect of Fungal Age on Protoplast Yield

As shown in Figure 2a, the protoplast yield of *L. decastes* increased initially with fungal age and then decreased. Under conditions of enzymatic digestion at a digestion temperature of 28 °C for a digestion time of 2.5 h, with magnesium sulfate as the osmotic stabilizer and an enzyme concentration of 1.75%, the optimal fungal age for protoplast preparation was found to be 10 days, yielding 4.305 × 10^6^ cells/mL. This yield was significantly higher than fungal ages of eight and 12 days. Consequently, nine, 10, and 11 days of fungal ages were selected for further orthogonal experiments.

#### 3.1.2. Effect of Digestion Time on Protoplast Yield

With an increasing digestion time, the protoplast yields initially increased. Under the conditions of a digestion temperature of 28 °C, a fungal age of 10 days, using MgSO_4_ as the osmotic stabilizer, and an enzyme concentration of 1.75%, the protoplast yield reached a maximum of 4.44 × 10^6^ cells/mL at a digestion time of 2.5 h (Figure 2b). However, when the digestion time exceeded 2.5 h, over-digestion led to protoplast rupture and a subsequent decrease in yield. Therefore, 2.25, 2.50, and 2.75 h digestion times were selected for further orthogonal experiments.

#### 3.1.3. Effect of Digestion Temperature on Protoplast Yield

Under the conditions of *L. decastes* strain at a fungal age of 10 days, using MgSO_4_ as the osmotic stabilizer, a digestion time of 2.5 h, and an enzyme concentration of 1.75%, the protoplast yield exhibited an initial increase, followed by a decrease as the digestion temperature increased within the range of 26–34 °C. The maximum protoplast yield of 4.20 × 10^6^ cells/mL was observed at a digestion temperature of 28 °C (Figure 2c). Therefore, digestion temperatures of 27 °C, 28 °C, and 29 °C were selected for further orthogonal experiments.

#### 3.1.4. Effect of Osmotic Stabilizers on Protoplast Yield

Under the conditions of a fungal age of 10 days, a digestion temperature of 28 °C, a digestion time of 2.5 h, and an enzyme concentration of 1.75%, the protoplast yield of *L. decastes* varied, depending on the osmotic stabilizer, as shown in Figure 2d. The yield followed the following order: MgSO_4_ > KCl > NaCl > sucrose > mannitol, with the highest yield being 4.095 × 10^6^ cells/mL. The performance of inorganic salts was generally superior to that of organic sugar alcohols.

#### 3.1.5. Effect of Enzyme Concentration on Protoplast Yield

Under the conditions of a fungal age of 10 days, using MgSO_4_ as the osmotic stabilizer, a digestion temperature of 28 °C, and a digestion time of 2.5 h, the effect of enzyme concentration on the protoplast yield was investigated, as shown in Figure 2e. With an increasing enzyme concentration, the protoplast yields initially increased and then decreased. At an enzyme concentration of 1.75%, the protoplast yield was 4.125 × 10^6^ cells/mL, while at a concentration of 2.0%, the yield reached a maximum of 4.395 × 10^6^ cells/mL. There was no significant difference between these yields. Therefore, enzyme concentrations of 1.5%, 1.75%, and 2.0% were selected for further orthogonal experiments.

### 3.2. Orthogonal Experiment for Protoplast Preparation of L. decastes Strain HMJU 6921

The single-factor experiment results indicated that fungal age, digestion time, enzyme concentration, and digestion temperature are key factors influencing the preparation of protoplasts from *L. decastes*. These factors often exhibit complex synergistic or inhibitory interactions, necessitating the use of orthogonal experiments to determine the optimal conditions for protoplast preparation (Table 2). The orthogonal experiment results (Table 4) showed that the main contributing factors to protoplast preparation, in descending order of influence, were fungal age (factor A), enzyme concentration (factor C), digestion temperature (factor D), and digestion time (factor B). The optimal conditions were determined to be as follows: a fungal age of 10 days, a digestion time of 2.25 h, an enzyme concentration of 2%, and a digestion temperature of 28 °C (A2B1C3D2). These optimal conditions corresponded to Trial 4, with a maximum protoplast yield of 5.475 × 10^6^ cells/mL.

### 3.3. Effect of Regeneration Media on Protoplast Regeneration of L. decastes Strain HMJU 6921

Protoplasts, having lost their cell walls, are more vulnerable to external environmental stimuli compared to mycelium. Therefore, the rapid and efficient regeneration and the proliferation of protoplasts are crucial for their effective utilization. In this study, five types of regeneration media conducive to fungal protoplast regeneration were developed based on the relevant literature (Table 3) to assess their regeneration rates. As shown in Figure 3, the regeneration rate of the PDMS medium, which uses sucrose as an osmotic stabilizer, was significantly higher than that of the PDAM medium, which uses mannitol as an osmotic stabilizer. However, PDMS exhibited weaker colony growth compared to PDAM, and both media had lower regeneration rates compared to the other three media. Regeneration medium Z3, which used sucrose as an osmotic stabilizer and was selected as the optimal medium from biological characteristic tests of the strain, resulted in a higher regeneration rate than PDMS and PDAM. Regeneration medium Z4, which added vitamin B to Z3, exhibited a slightly lower regeneration rate than Z3, but the difference was not statistically significant. Regeneration medium Z5, which incorporated *L. decastes* mycelium extract (prepared by drying approximately 80 g of mycelium, boiling in 1000 mL of water, and concentrating to 200 mL), exhibited a significantly higher regeneration rate than the other four media and demonstrated robust colony growth. The regeneration rate could reach 2.86%. Media that effectively support protoplast regeneration not only enhance the success rate of subsequent experiments but also reduce the fungal growth time, thereby accelerating the overall experimental process.

### 3.4. Overview of Transcriptome Sequencing Data Analysis

RNA-seq was conducted during three distinct stages of protoplast preparation and regeneration, resulting in three samples, each with three biological replicates, generating a total of nine cDNA libraries: LDY1_1, LDY1_2, LDY1_3, LDY2_1, LDY2_2, LDY2_3, LDY3_1, LDY3_2, and LDY3_3. The RNA quality test results are shown in Appendix A, and the gel image is shown in Appendix A. The raw sequencing data were deposited in the NCBI database under the accession number PRJNA1187305. Each sample generated approximately 50,678,112 to 70,480,936 raw reads, with Q20 and Q30 percentages exceeding 98.93% and 96.89%, respectively. In total, 33.44 G of raw data was obtained. After adapter sequences and low-quality reads were filtered out, each sample yielded more than 50,007,930 clean reads. The clean reads were assembled using Trinity software (v2.15.1), resulting in a total unigene length of 33,124,310 bp, with an average length of 1498.97 bp and a GC content of 51.00%. All assembled unigenes and transcripts were compared to six major databases (Nr, Swiss-Prot, Pfam, COG, GO, and KEGG) to annotate their functions (Figure 4a).

### 3.5. Analysis of Differential Gene Expression During Protoplast Preparation and Regeneration

Differentially expressed genes (DEGs) during protoplast preparation and regeneration were identified using thresholds of |log2FoldChange| > 1 and *p*-value < 0.05. A total of 13,120 genes were found to be differentially expressed during the protoplast preparation and regeneration stages (Figure 4b). Specifically, 2432 DEGs (1195 upregulated and 1237 downregulated) were observed in the comparison between LDY1 and LDY2, 4836 DEGs (3016 upregulated and 1757 downregulated) in LDY2 vs. LDY3, and 5852 DEGs (3827 upregulated and 2025 downregulated) in LDY1 vs. LDY3.

### 3.6. Functional Annotation of Differentially Expressed Genes During Protoplast Preparation and Regeneration

Gene ontology (GO) includes three categories: molecular function (MF), cellular component (CC), and biological process (BP). The GO analysis during protoplast preparation and regeneration is shown in Figure 5. The results showed that DEGs participate in multiple biologically significant processes. A GO enrichment analysis for DEGs between protoplasts and the mycelial stage (LDY2 vs. LDY1) identified 51 CC, 92 MF, and 352 BP terms (Figure 5a). Enriched CC terms included cytosolic ribosome, ribosome, ribosomal subunit, cytosolic large ribosomal subunit, peroxisome, and microbody, associated with protein synthesis and metabolic regulation. Enriched MF terms included structural constituent of ribosome, structural molecule activity, aspartic-type endopeptidase activity, RNA-directed DNA polymerase activity, and oxidoreductase activity, linked to ribosomal structure and oxidative stress responses. BP terms included cytoplasmic translation, translation, peptide biosynthetic process, cellular response to oxidative stress, and response to oxidative stress, highlighting involvement in protein synthesis and oxidative stress. For DEGs between the protoplast regeneration and protoplast stages (LDY2 vs. LDY3), 110 CC, 110 MF, and 407 BP terms were identified (Figure 5b). Enriched CC terms included cytosolic ribosome, cytosolic large ribosomal subunit, ribosome, cytosol, and cytoplasm, essential for protein synthesis and metabolic activity. Enriched MF terms included structural constituent of ribosome, aspartic-type endopeptidase activity, unfolded protein binding, antioxidant activity, and peroxidase activity, reflecting roles in protein structure, degradation, and antioxidative responses. BP terms included cytoplasmic translation, protein folding, purine ribonucleoside triphosphate metabolic process, drug metabolic process, and oxidative phosphorylation, indicating protein synthesis, metabolic reorganization, and energy production. For DEGs between regenerated mycelium and mycelium before cell wall removal (LDY1 vs. LDY3), 107 CC, 128 MF, and 385 BP terms were identified (Figure 5c). Enriched CC terms included cytosolic ribosome, cytosolic large ribosomal subunit, ribosomal subunit, ribosome, and cytosol, highlighting protein synthesis in the cytoplasm. Enriched MF terms included structural constituent of ribosome, unfolded protein binding, proton channel activity, proton-transporting ATP synthase activity, and oxidoreductase activity, reflecting roles in protein synthesis, energy metabolism, and antioxidant defense. BP terms included cytoplasmic translation, peptide biosynthetic process, protein folding, cell redox homeostasis, and chaperone-mediated protein folding, highlighting protein synthesis, folding, and redox balance. Shared GO terms during protoplast preparation and regeneration (LDY1 vs. LDY2 and LDY2 vs. LDY3) showed a significant enrichment of ribosome-related CC terms, including cytosolic ribosome, cytosolic large ribosomal subunit, ribosomal subunit, and cytosol. Enriched MF terms included structural constituent of ribosome, structural molecule activity, aspartic-type endopeptidase activity, and aspartic-type peptidase activity. The only shared BP term was cytoplasmic translation. These common GO terms suggest that ribosomes are highly active during protoplast preparation and regeneration, with protein synthesis and quality control likely serving as core activities.

To further elucidate the biological pathways associated with these DEGs, we conducted KEGG enrichment analysis for comparisons between LDY1 vs. LDY2, LDY2 vs. LDY3, and LDY1 vs. LDY3, selecting the top 30 enriched pathways (Figure 6). In the KEGG enrichment analysis of DEGs between LDY1 and LDY2 (Figure 6a), the most significantly enriched pathways included ribosome, peroxisome, fatty acid degradation, beta-alanine metabolism, the biosynthesis of unsaturated fatty acids, histidine metabolism, glycolysis/gluconeogenesis, steroid biosynthesis, and tryptophan metabolism. These enriched pathways indicate that, during the removal of the fungal cell wall to form protoplasts, cells may adapt to structural changes and metabolic stress by regulating protein synthesis and energy metabolism, as well as lipid and amino acid metabolic activities. In the LDY2 vs. LDY3 stage (Figure 6b), the primary enriched pathways included ribosome, oxidative phosphorylation, proteasome, glutathione metabolism, peroxisome, various types of N-glycan biosynthesis, arginine and proline metabolism, glycosphingolipid biosynthesis—Globo and Isoglobo series, and protein processing in the endoplasmic reticulum. The enrichment of these pathways suggests that, during the regeneration phase, protoplasts may maintain physiological homeostasis and ensure successful regeneration through the coordinated regulation of protein synthesis, energy metabolism, protein degradation, and antioxidant mechanisms. Similarly, a KEGG enrichment analysis of DEGs between LDY1 and LDY3 (Figure 6c) identified pathways including ribosome, proteasome, glycolysis/gluconeogenesis, steroid biosynthesis, pyruvate metabolism, oxidative phosphorylation, seleno compound metabolism, arginine and proline metabolism, glutathione metabolism, and endocytosis. This indicates that regenerated mycelia maintain high metabolic activity related to protein metabolism, energy generation, and antioxidant stress, ensuring functional recovery and physiological homeostasis. Additionally, by analyzing KEGG pathways shared between protoplast preparation and regeneration (LDY1 vs. LDY2 and LDY2 vs. LDY3), we found the consistent enrichment of pathways such as ribosome, oxidative phosphorylation, peroxisome, arginine and proline metabolism, glycolysis/gluconeogenesis, pyruvate metabolism, fatty acid degradation, fatty acid biosynthesis, taurine and hypotaurine metabolism, pantothenate and CoA biosynthesis, beta-alanine metabolism, alpha-linolenic acid metabolism, glyoxylate and dicarboxylate metabolism, and the biosynthesis of unsaturated fatty acids. This suggests that, whether during fungal cell wall removal to form protoplasts or during protoplast regeneration, protein synthesis and antioxidant stress responses may play critical roles.

### 3.7. Validation of RNA-Seq Data via qRT-PCR

In order to confirm the transcriptome data, qRT-PCR was performed to examine the ex-pression patterns of genes associated with protoplast preparation and regeneration. The qRT-PCR results aligned with the RNA-Seq findings, demonstrating the reliability of our RNA-Seq data through the concordance of the two methods (Figure 7a).

### 3.8. Genes and Proteins Potentially Critical for Protoplast Preparation and Regeneration

An analysis of differentially expressed genes during protoplast preparation and regeneration (Figure 7b) revealed that, in the protoplast formation stage, compared with unstripped mycelium (LDY1 vs. LDY2), genes encoding Casein kinase I, peroxisome-associated protein, Cytochrome P450 family, and oxidative stress response two-component system protein were significantly upregulated. Notably, the genes encoding casein kinase I showed substantial upregulation, with fold changes of 9.77 (HHP1) and 34.34 (CK1). Casein kinase I has been shown to be essential for resistance to cell membrane stress in *Candida* [26] and crucial for maintaining cell membrane and cell wall integrity in *Cryptococcus neoformans* [27]. In the differentially expressed genes between regenerated mycelium and protoplasts (LDY3 vs. LDY2), genes involved in encoding chitin synthase, beta-glucan synthesis-associated protein, cell wall alpha-1,3-glucan synthase, and hydrophobin protein were all upregulated. Chitin, β-glucan, and α-1,3-glucan are key components of fungal cell walls. Additionally, five genes involved in encoding hydrophobin showed significant upregulation. Hydrophobin plays multiple roles in supporting fungal growth and development [28], and it is essential for the formation of aerial hyphae in fungi [29,30,31]. Furthermore, in the differentially expressed genes between regenerated mycelium and unstripped mycelium (LDY1 vs. LDY3), heat shock protein was downregulated, while septin protein, 1,3-beta-glucan synthase component, and another 1,3-beta-glucan synthase component were upregulated. Heat shock protein is known to be highly induced under stress conditions [32,33,34]. Septins are evolutionarily conserved GTP-binding proteins that form complexes and participate in various essential cellular processes, including cytokinesis [35,36], phagocytosis [37], ciliogenesis [38], cell polarization [39], and morphogenesis [40,41].

### 3.9. Arginine and Proline Metabolism Pathway

Arginine and proline play multiple essential roles in plants and microorganisms [42], including assisting cells in coping with oxidative stress, regulating osmotic pressure [43], mobilizing nitrogen reserves through catabolism, fine-tuning development, and enhancing defense mechanisms against environmental stress [44]. KEGG enrichment analysis revealed that this metabolic pathway was enriched during three stages: protoplast preparation (LDY1 vs. LDY2), regeneration (LDY2 vs. LDY3), and the stage between regenerated mycelium and pre-cell-wall-removal mycelium (LDY1 vs. LDY3) (Figure 8). As shown in Figure 9, during the LDY1 vs. LDY2 stage, one gene was upregulated, and nine genes were downregulated; during the LDY2 vs. LDY3 stage, 19 genes were upregulated, and six were downregulated, and in the LDY1 vs. LDY3 stage, 12 genes were upregulated, and 10 were downregulated. These findings suggest that this pathway plays an important role in both protoplast formation and the recovery of regenerated mycelium.

## 4. Discussion

In the preparation of fungal protoplasts, the primary influencing factors include enzyme concentration, osmotic stabilizers, temperature, enzyme digestion time, and fungal age [18,19,20]. The optimal conditions for protoplast preparation vary among different fungal species [45]. In protoplast preparation, the cell wall is the substrate for enzymatic degradation. The physiological state of the mycelium directly affects the structure of the cell wall and mycelial vitality. Among them, the fungal age has a particularly significant effect on the generation of protoplasts. The cell walls of newborn mycelia are thin and easy to degrade, but there are few mycelia, and the yield is low. The cell walls of older mycelia are thicker and difficult to completely enzymolyze. Sometimes, the enzymolysis products only contain mycelial fragments. This study used mycelia cultured for 10 d of mycelium to prepare protoplasts, which yielded a higher yield. Compared with other studies [46], the fungal age selected in this experiment was relatively long. On the one hand, this was due to the slow growth rate of *L. decastes* itself, which also varies among different strains of the same species [47]. On the other hand, this strain was obtained from a wild strain via tissue separation, and it grows slowly without acclimatization. The enzymatic depelling of the mycelium is consistent with the kinetic principles of enzymatic reactions; i.e., within a certain range, as the amount of enzyme increases, the yield of protoplasts increases, but beyond a certain range, the rate of enzymatic reaction will not increase significantly. Furthermore, enzymes inevitably contain nucleases and proteases. Excessive enzyme concentration or prolonged enzymatic hydrolysis can damage the cell membrane glycoproteins and nucleic acids of protoplasts, reducing their activity and making it difficult to restore the cell structure [48]. Similarly, other conditions of enzymatic digestion, such as temperature and time, are only optimal when the conditions are most suitable. The enzyme activity is highest at lower temperatures, and the protoplasts form more slowly. At higher temperatures, it is necessary to consider whether the strain can withstand high temperatures. The optimal culture temperature for *L. decastes* is 20–25 °C [47]. The digestion time is short, and there is little protoplast release. A long digestion time can destroy the membrane structure of the protoplast. Different osmotic stabilizers have a considerable effect on the protoplast yield [45]. The protoplasts, which have had their cell walls removed, are fragile and need to be suspended in a specific osmotic stabilizer to maintain their biological activity and prevent cell lysis. Osmotic stabilizers are divided into two categories: organic and inorganic. In this experiment, the preparation effect of inorganic salt osmotic stabilizers was significantly higher than that of organic sugar alcohols. Among them, magnesium sulfate had the best preparation effect, and mg^+^ had a promoting effect on most enzymes [49], while sucrose and mannitol had an inhibitory effect on alpha glucanase [50,51], and alpha-glucan is one of the components of the fungal cell wall [49]. This experiment optimized the single-factor test and orthogonal test for the above factors affecting protoplast preparation and found that *L. decastes* mycelium at a fungal age of 10 d, an enzyme concentration of 2.0%, a digestion time of 2.25 h, a digestion temperature of 28 °C, and 0.6 mol/L magnesium sulfate as an osmotic pressure stabilizer, can more fully enzymolyze the cell wall, and a high protoplast yield, reaching 5.475 × 10^6^ cells/mL. The regeneration of protoplasts includes the regeneration and restoration of cell walls, that is, the restoration of hyphal germination and growth [52]. In the optimization of protoplast regeneration conditions, the selection of the regeneration medium is crucial. We screened five different regeneration media and found that they had a significant effect on the regeneration rate of protoplasts. In this study, the most suitable carbon source, nitrogen source, and inorganic salt type for the growth of this strain were obtained in advance, and the Z3 medium was obtained based on the PDMS medium. All regeneration media contained a large amount of a carbon source, which meant that the carbon source was not the key point for protoplast regeneration. The regeneration rates of Z3, Z4, and Z5 media, which had additional nitrogen sources added, were higher than those of the other two media. The main function of the nitrogen source is protein synthesis, which plays a very important role in microbial proliferation. The nitrogen source may have played a role in promoting cell division during the regeneration of *L. decastes* protoplasts, causing the mycelium to proliferate rapidly. Some studies have shown that adding a certain concentration of cell wall precursor substances and nutrients such as mycelium leachates and vitamin B1 to the regeneration medium can increase the protoplast regeneration rate [53] (p. 72). Medium Z4 contains a complex vitamin B supplement, and the protoplast yield is slightly lower than that of medium Z3, but there is no significant difference. Z5 medium, which contains the extract of *L. decastes*, had the highest regeneration rate of 2.87%, which is higher than in previous studies [46].

The regeneration rate of mushroom protoplasts is generally low [53] (p. 115). In a previous article, we established an efficient protoplast preparation method by optimizing the preparation and regeneration conditions, laying the foundation for subsequent transcriptomics research. To further reveal the molecular mechanism in the preparation and regeneration process, we analyzed the transcription levels of three key stages: undecapped mycelium (LDY1), protoplasts (LDY2), and regenerated mycelium (LDY3). In the differential gene pathway enrichment analysis of LDY1 vs. LDY2, we found that a large number of genes in pathways such as ribosome (ko03010), histidine metabolism (ko00340), glycolysis/gluconeogenesis (ko00010), and arginine and proline metabolism (ko00330) were downregulated, while pathways such as peroxisome (ko04146), fatty acid degradation (ko00071), and oxidative phosphorylation (ko00190) were upregulated. The downregulation of the ribosome pathway indicates that protoplasts may temporarily reduce the synthesis of non-essential proteins and that ribosome synthesis and assembly require a lot of energy [54]. Cells may prioritize limited energy for maintaining basic survival. At the same time, the downregulation of non-essential amino acid metabolism such as histidine, arginine, and proline represents a strategic redistribution of amino acid metabolism by cells. It is particularly noteworthy that the energy metabolism pattern may shift from relatively inefficient glycolysis (only two ATPs per glucose molecule) to fatty acid degradation and oxidative phosphorylation (30–36 ATPs). This shift reflects the strategy of protoplasts to pursue more efficient energy production. There are also multiple antioxidant systems in peroxisomes [55]. The upregulation of this pathway indicates that cells may enhance their antioxidant defenses. At the same time, the activation of the fatty acid degradation pathway not only provides protoplasts with an alternative energy source but also supports the stability of the cell membrane through metabolites [56]. The enhancement of oxidative phosphorylation ensures the energy supply efficiency of protoplasts under stress [57]. In general, the metabolic characteristics of the protoplast period changed. Among them, the pathways related to protein synthesis (ribosome), non-essential amino acid metabolism (histidine, arginine, and proline metabolism), and glucose metabolism (glycolysis/gluconeogenesis) were downregulated, indicating that protoplasts reduced energy consumption and reduced growth metabolic activity; at the same time, energy metabolism and stress response-related pathways such as peroxisome (peroxisome), fatty acid degradation (fatty acid degradation), and oxidative phosphorylation (oxidative phosphorylation) were upregulated, indicating that protoplasts may be undergoing a metabolic mode change from growth-oriented metabolism to a metabolic mode dominated by maintaining survival and coping with stress. This change may be an important mechanism for protoplasts to adapt to the environment after losing the cell wall. In addition, during this period, we found that casein kinase I, peroxisome-associated protein, cytochrome P450, and oxidative stress response two-component system protein were all upregulated. In *Cryptococcus neoformans*, casein kinase I protein Cck1 is important for maintaining cell membrane integrity, and it affects the cell’s response to osmotic pressure through the HOG pathway [27]. The cytochrome P450 (CYP) gene family is widely distributed in fungi and plays an important role in regulating fungal responses to abiotic stress [58,59]. The CYP gene family mainly regulates fungal responses to abiotic stress by participating in various physiological responses. Among them, CYP52 protein catalyzes the conversion of fatty acids and alkanes into α, omega-dicarboxylic acids [60], which are the main structural components of membrane lipids in *Thermoanaerobacter ethanolicus* [61]. CYP56 (TRINITY_DN482_c4_g1) can catalyze the conversion of N-formyltyrosine to N, N-diformyldityrosine, generating dityrosine cross-links. This cross-link helps enhance the rigidity and stability of the fungal cell wall. By regulating the repair of the cell wall, CYP56 indirectly affects the integrity of the cell membrane [62]. PEX5 acts as a redox-regulated input receptor and plays a role in cellular defense, counteracting oxidative damage from peroxisomes in vitro [63]. In *Candida auris*, Ssk1 mainly acts as a key regulator in the HOG signaling pathway. Oxidative stress response two-component system protein SSK1-dependently activates Hog1, which provides protection against oxidative stress, indicating that Ssk1 plays an important role in regulating the cellular response to oxidative stress [64,65]. The upregulated expression of these genes suggests that protoplasts are actively responding to environmental stress and maintaining their homeostasis by enhancing signal transduction, metabolic regulation, and antioxidant capacity.

In the differential gene functional enrichment analysis during the LDY2 vs. LDY3 period, the upregulation of multiple key pathways reveals biological activities in the regeneration process. First, the enhancement of protein synthesis and processing (such as ribosome and protein processing in the endoplasmic reticulum) indicates that the cells require rapid synthesis and processing of a large number of proteins to support hyphal regeneration. Second, active energy metabolism (such as oxidative phosphorylation and glycolysis/gluconeogenesis) reflects a high demand for energy during the regeneration process. The upregulation of antioxidant and stress responses (such as glutathione metabolism and peroxisome) suggests that the regenerated hyphae may possess strong protective capabilities to cope with oxidative stress. In particular, 23 of the 30 DEGs annotated to the glutathione metabolic pathway were upregulated. Glutathione, a tripeptide thiol present in almost all cells, is an important metabolite in eukaryotes, and it plays an important role in the metabolism, transport, and protection of cells from oxidative damage [66,67,68]. Additionally, the activation of metabolic regulation and biosynthesis (such as arginine and proline metabolism, various types of N-glycan biosynthesis, and glycosphingolipid biosynthesis) ensures that the cells acquire the necessary nutrients and biosynthetic capabilities. Finally, the upregulation of RNA monitoring and quality control (such as surveillance pathways and proteasome) emphasizes the importance of gene expression and protein quality during the regeneration process. The synergistic action of these pathways may reflect the significant physiological adaptations of the regenerated hyphae compared to protoplasts, providing a reference for a deeper understanding of their regeneration mechanisms. Moreover, we also observed the upregulation of beta-glucan synthesis-associated protein (SKN1), chitin synthase 2 (CHS2), hydrophobin-1 (HYP1), and hydrophobin-2 (HYP2). The fungal cell wall is primarily composed of glucans, chitin, and mannoproteins [69]. SKN1, as a beta-glucan synthesis-related protein, mainly participates in the construction of the yeast cell wall by influencing sphingolipid biosynthesis [70]. CHS2 is a chitin synthase specifically responsible for primary septum formation in yeast cells, which is crucial for maintaining normal cell morphology, size, and division. Its deletion leads to severe cell morphology abnormalities and septum structure defects. As mentioned earlier, the regeneration of protoplasts involves both cell wall regeneration and the resumption of hyphal growth. After completing cell wall repair, protoplasts typically grow hyphae via apical extension. The hyphae penetrate the moist substrate, establish branching internal hyphae, and then differentiate to grow from the substrate into the air, forming aerial hyphae, with hydrophobins being key to the formation of aerial hyphae [31]. Additionally, the cell wall alpha-1,3-glucan synthase Ags 1 (TRINITY_DN1744_c1_g1) has been demonstrated to be essential for the synthesis of 1,3-alpha-glucan in the aerial hyphae of rough pathogen [71].

Finally, we analyzed the enrichment of differentially expressed genes between the LDY1 and LDY3 stages. Interestingly, there were significant metabolic characteristics between the regenerated and controlled mycelia, with numerous pathways upregulated. These pathways could be categorized into several functional groups: (1) pathways related to protein metabolism, including ribosome (ko03010), proteasome (ko03050), protein export (ko03060), and endoplasmic reticulum protein processing (ko04141), indicating enhanced protein synthesis, processing, and turnover; (2) energy metabolism pathways, including glycolysis/gluconeogenesis (ko00010) and oxidative phosphorylation (ko00190), reflecting increased energy demand; (3) amino acid and nitrogen metabolism pathways, including arginine and proline metabolism (ko00330), selenium compound metabolism (ko00450), and nitrogen metabolism (ko00910), suggesting active turnover of nitrogen-containing compounds; (4) transport and cellular processes, such as ABC transporters (ko02010) and endocytosis (ko04144), indicating enhanced material exchange; and (5) stress response pathways, like glutathione metabolism (ko00480). Notably, one pathway showed significant downregulation: steroid biosynthesis (ko00100). The downregulation of steroid biosynthesis may suggest a reduced need for membrane modification in the later stages of regeneration. Collectively, these metabolic features indicate that, although both states maintain intact cell walls, the protoplast’s dedifferentiation and redifferentiation process may trigger the reprogramming of cell growth and development, as evidenced by the activation of multiple biosynthesis pathways (e.g., ribosome and protein processing). Similarly, at this stage, we also selected differentially expressed genes for qRT-PCR to validate the accuracy of transcriptome data, finding that the upregulated expression of chitin synthase 1 and 1,3-beta-glucan synthase component indicates the regenerated mycelium’s strong demand for cell wall synthesis, environmental adaptation, and growth. Meanwhile, the downregulation of heat shock protein expression suggests that the regenerated mycelium may be undergoing rapid growth or differentiation. During this phase, the expression of heat shock proteins may be suppressed via the activation of other growth-related genes, promoting cell growth and functional recovery.

## 5. Conclusions

In this study, we optimized the conditions for protoplast preparation and regeneration in *L. decastes* and analyzed gene expression changes at different preparation stages using transcriptomics. One-way experiments identified key factors affecting protoplast preparation in strain HMJU 6921, including the type of osmotic stabilizer, enzyme concentration, digestion temperature, digestion time, and mycelial age. Orthogonal experiments determined the optimal conditions: 10-day-old mycelia, 2.25 h of digestion, a 2% enzyme concentration, and a 28 °C digestion temperature, yielding 5.475 × 10^6^ cells/mL of protoplasts. The regeneration rate on medium Z5 supplemented with *L. decastes* mycelial extract reached 2.86%. Transcriptomic analysis revealed significant changes in key biological pathways and genes related to cell wall synthesis, stress responses, and metabolic regulation during protoplast preparation and regeneration. This study provides theoretical and technical support for efficient *L. decastes* protoplast preparation and regeneration, offering valuable insights for further research, including genetic modification and breeding, and advancing the application potential of *L. decastes*.

## Figures and Tables

**Figure 1 jof-10-00886-f001:**
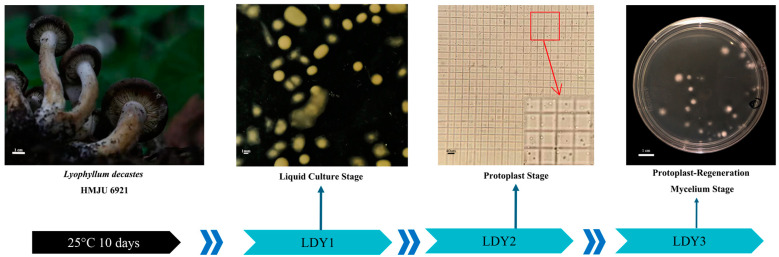
Sampling periods and processed samples of *L. decastes*.

**Figure 2 jof-10-00886-f002:**
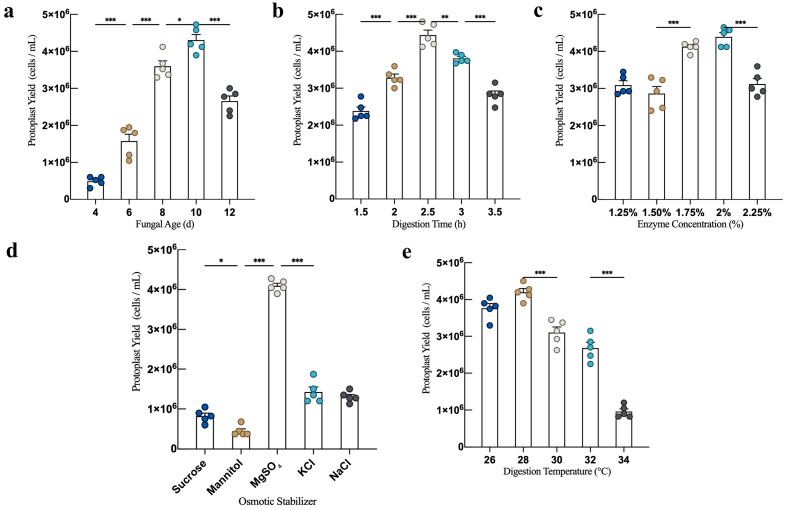
Optimization of protoplast preparation conditions for *L. decastes* strain HMJU 6921. (**a**) fungal age. (**b**) Digestion time. (**c**) Digestion temperature. (**d**) Osmotic stabilizer. (**e**) Enzyme concentration. (* *p* < 0.033, ** *p* < 0.002, and *** *p* < 0.001).

**Figure 3 jof-10-00886-f003:**
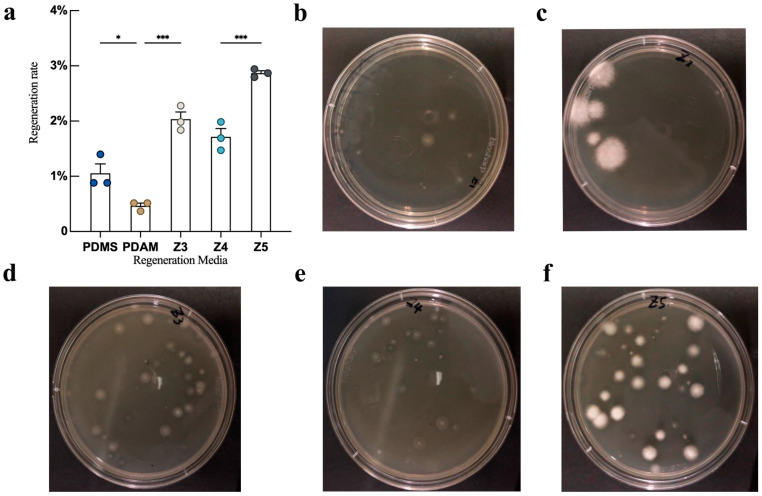
Regeneration of *L. decastes* strain HMJU 6921 in different regeneration media. (**a**) Regeneration rate of *L. decastes* strain HMJU 6921 in different regeneration media. (**b**) PDMS. (**c**) PDAM. (**d**) Z3. (**e**) Z4. (**f**) Z5. (Circles of the same color represent the number of replicates in the experiment, * *p* < 0.033, and *** *p* < 0.001).

**Figure 4 jof-10-00886-f004:**
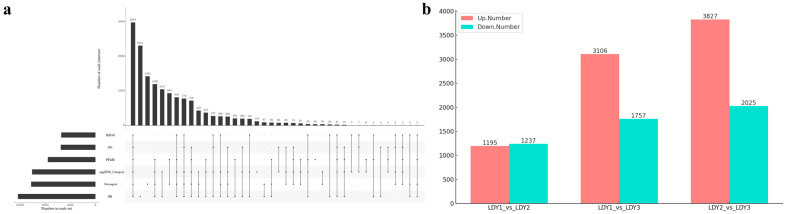
Transcriptome analysis of *L. decastes.* (**a**) Upset plot of unigene annotation statistics. (**b**) Numbers of differentially expressed genes compared between two samples.

**Figure 5 jof-10-00886-f005:**
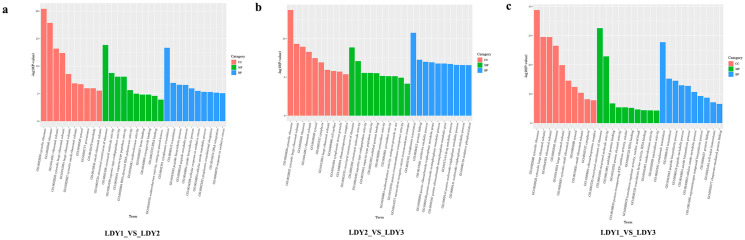
GO enrichment analysis of DEGs during protoplast preparation and regeneration. (**a**) LDY1 vs. LDY2. (**b**) LDY2 vs. LDY3. (**c**) LDY1 vs. LDY3.

**Figure 6 jof-10-00886-f006:**
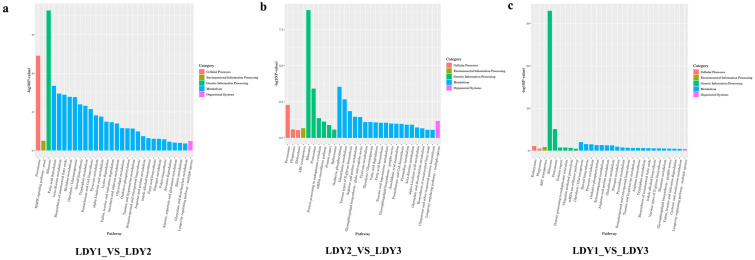
KEGG pathway enrichment analysis of DEGs during protoplast preparation and regeneration. (**a**) LDY1 vs. LDY2. (**b**) LDY2 vs. LDY3. (**c**) LDY1 vs. LDY3.

**Figure 7 jof-10-00886-f007:**
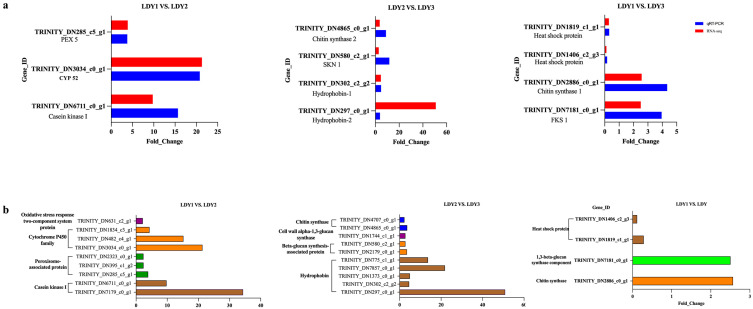
qRT-PCR results and expression profiles of genes related to protoplast preparation and regeneration. (**a**) Expression profiles of genes related to protoplast preparation and regeneration were validated via a qRT-PCR analysis. (**b**) Expression profiles of genes potentially involved in protoplast preparation and regeneration processes.

**Figure 8 jof-10-00886-f008:**
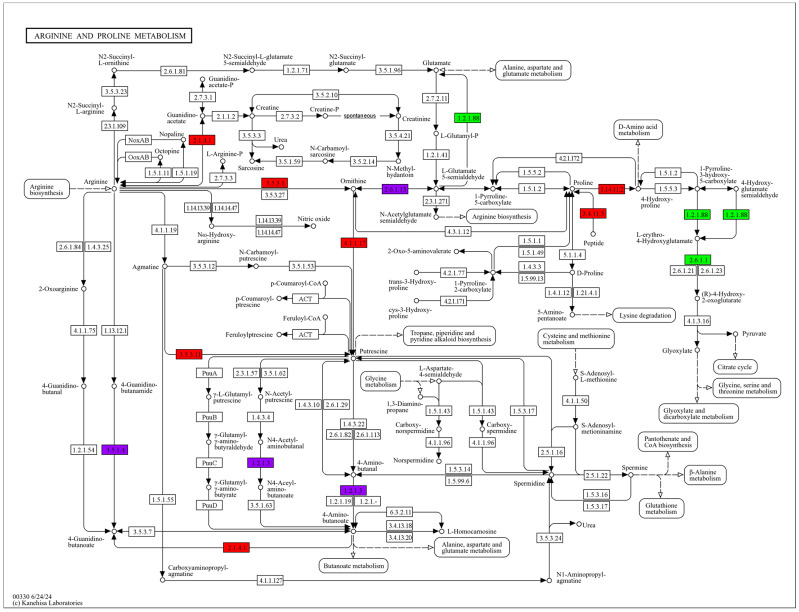
Differentially expressed genes in LDY1 vs. LDY2, LDY2 vs. LDY3, and LDY1 vs. LDY3 were mapped to the arginine and proline metabolism pathway (ko00330). Red indicates upregulation, green indicates downregulation, and purple indicates both upregulation and downregulation.

**Figure 9 jof-10-00886-f009:**
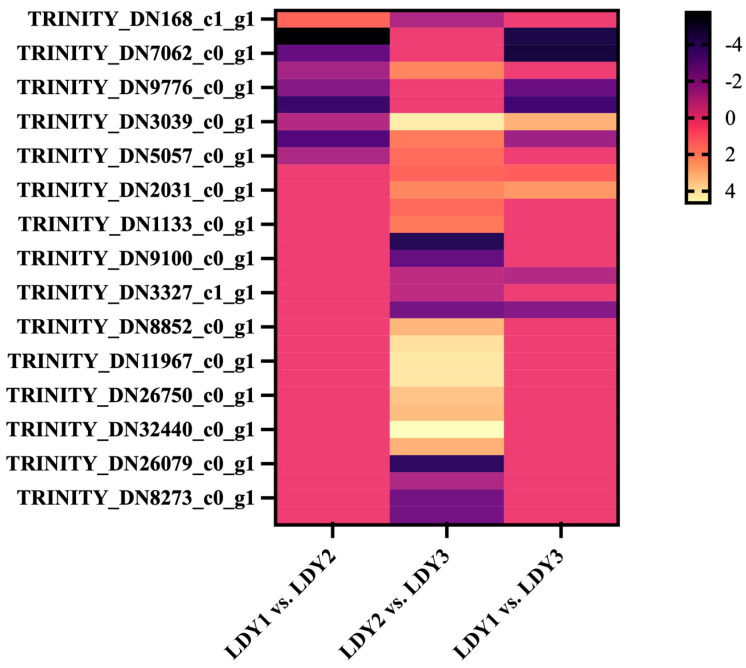
Heatmap of differentially expressed genes involved in the arginine and proline metabolism pathway (ko00330), with red indicating upregulated expression and blue indicating downregulated expression.

**Table 1 jof-10-00886-t001:** Single-factor experimental design for protoplast preparation of *L. decastes*.

Factor	Level 1	Level 2	Level 3	Level 4	Level 5
Fungal Age	4 days	6 days	8 days	10 days	12 days
Digestion Time	1.5 h	2 h	2.5 h	3 h	3.5 h
Digestion Temperature	26 °C	28 °C	30 °C	32 °C	34 °C
Osmotic Stabilizer	Sucrose	Mannitol	MgSO_4_	NaCl	KCl
Enzyme Concentration	1.25%	1.50%	1.75%	2%	2.25%

**Table 2 jof-10-00886-t002:** Orthogonal experimental design for protoplast preparation of *L. decastes*.

Factor	Level 1	Level 2	Level 3
A: Fungal Age	9 days	10 days	11 days
B: Digestion Time	2.25 h	2.5 h	2.75 h
C: Enzyme Concentration	1.50%	1.75%	2%
D: Digestion Temperature	27 °C	28 °C	29 °C

**Table 3 jof-10-00886-t003:** Regeneration media components.

Regeneration Medium	Components	Reference
PDMS	Potato, 200 g/L; glucose, 20 g/L; maltose, 1 g/L; sucrose, 0.6 mol/L; agar, 20 g/L; water, 1 L	[21]
PDAM	Potato, 200 g/L; glucose, 20 g/L; mannitol, 0.6 mol/L; agar, 20 g/L; water, 1 L	[22]
Z3	Potato, 200 g/L; starch, 20 g/L; NaCl, 2 g/L; peptone, 2 g/L; sucrose, 0.6 mol/L; agar, 20 g/L; water, 1 L	This study
Z4	Potato, 200 g/L; starch, 20 g/L; NaCl, 2 g/L; peptone, 2 g/L; sucrose, 0.6 mol/L; vitamin B, 10 mg/L; agar, 20 g/L; water, 1 L	This study
Z5	Potato, 200 g/L; starch, 20 g/L; NaCl, 2 g/L; peptone, 2 g/L; sucrose, 0.6 mol/L; 200 mL of *L. decastes* mycelium extract; agar, 20 g/L; water, 1 L	This study

**Table 4 jof-10-00886-t004:** Orthogonal experiment results for protoplast preparation of *L. decastes*.

No.	A	B	C	D	Protoplast Yield (10^6^ Cells/mL)
	Fungal Age	Digestion Time	Enzyme Concentration	Digestion Temperature	
1	9 d	2.75 h	2%	29 °C	2.685 ± 0.194
2	9 d	2.25 h	1.5%	27 °C	2.07 ± 0.173
3	9 d	2.5 h	1.75%	28 °C	4.455 ± 0.303
4	10 d	2.25 h	2%	28 °C	5.475 ± 0.29
5	10 d	2.75 h	1.75%	27 °C	3.585 ± 0.478
6	10 d	2.5 h	1.5%	29 °C	2.925 ± 0.428
7	11 d	2.5 h	2%	27 °C	2.52 ± 0.196
8	11 d	2.75 h	1.5%	28 °C	1.845 ± 0.203
9	11 d	2.25 h	1.75%	29 °C	2.43 ± 0.325
K1	9.21	9.97	6.84	8.17	
K2	11.98	9.9	10.47	11.78	
K3	6.79	8.12	10.68	8.04	
R	1.73	0.62	1.28	1.25	

## Data Availability

The transcriptome data presented in this study have been deposited in the National Center for Biotechnology Information Sequence Read Archive under accession number PRJNA1187305: https://dataview.ncbi.nlm.nih.gov/object/PRJNA1187305?reviewer=hd46e7cioka86767eivpk3p5s3 (accessed on 18 November 2024).

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
