# Peer review of "Optimization of Protoplast Preparation Conditions in Lyophyllum decastes and Transcriptomic Analysis Throughout the Process"

_jof, 2024, doi:10.3390/jof10120886_

Round 1
Reviewer 1 Report
The conditions of protoplast preparation in Lyophyllum decastes and transcriptome analysis in dynamics were investigated in the work. Some comments.
It is necessary to edit subsection 3.6 in the Results section.
It is necessary to write the Conclusion section more clearly and briefly.
It is necessary to correct the signature in Fig. 8.
It is necessary to provide a separate subsection on the statistical method in the Materials and Methods section.
It is necessary to illustrate the content of ribosomes in the cell during the preparation of protoplasts using the electron microscopic method.
It is desirable to provide the cellular response to oxidative stress by ROS production.
It is desirable to describe glutathione metabolism in more detail.
It is necessary to bring the list of references in line with the requirements of the journal.
Line 4,
fig.8
Author Response
Thank you for your comments concerning our manuscript entitled “Optimization of Protoplast Preparation Conditions in Lyophyllum decastes and Transcriptomic Analysis Throughout the Process”. Those comments were all valuable and very helpful in revising and improving our manuscript. We have studied the comments carefully and have made corrections which we hope meet with your approval. Revised portions are indicated in red. The main corrections in the paper and the response to the reviewer’s comments are as follows:
Comments 1: It is necessary to edit subsection 3.6 in the Results section.
Response 1: Thank you for pointing this out. We have re-edited subsection 3.6 of the Results section by removing redundant expressions, rearranging sentence structures, replacing repetitive content with abbreviations, and omitting GO and KEGG identifiers to present the findings more concisely and clearly.
Page 10-12, paragraph 3.6,Line 302-405:Change “The Gene Ontology (GO) analysis during protoplast preparation and regeneration is shown in Figure 5. GO includes three ontologies that describe the molecular function, cellular component, and biological process of genes. To determine the potential contributions of differentially expressed genes (DEGs), we conducted a GO functional enrichment analysis. The results indicated that DEGs are involved in several biologically significant activities. GO enrichment analysis of DEGs between protoplasts, formed by dissolving the mycelial cell wall and the mycelial stage (LDY2 vs. LDY1), identified 51, 92, and 352 GO terms related to cellular components, molecular functions, and biological processes, respectively (Figure 5a). In the cellular component (CC) category, significantly enriched terms included Cytosolic Ribosome (GO:0022626), Ribosome (GO:0005840), Ribosomal Subunit (GO:0044391), Cytosolic Large Ribosomal Subunit (GO:0022625), Peroxisome (GO:0005777), and Microbody (GO:0042579), which are associated with protein synthesis and metabolic regulation. In the molecular function (MF) category, enriched terms included Structural Constituent of Ribosome (GO:0003735), Structural Molecule Activity (GO:0005198), Aspartic-Type Endopeptidase Activity (GO:0004190), RNA-Directed DNA Polymerase Activity (GO:0003964), and Oxidoreductase Activity (GO:0016491), which are primarily related to ribosomal structure and oxidative stress responses. In the biological process (BP) category, significantly enriched terms included Cytoplasmic Translation (GO:0002181), Translation (GO:0006412), Peptide Biosynthetic Process (GO:0043043), Cellular Response to Oxidative Stress (GO:0034599), and Response to Oxidative Stress (GO:0006979), indicating involvement in protein synthesis and oxidative stress responses. The GO analysis of differentially expressed genes (DEGs) between the protoplast regeneration stage and the protoplast stage (LDY2 vs. LDY3) identified 407 terms related to biological processes, 110 to cellular components, and 110 to molecular functions (Figure 5b). In the cellular component (CC) category, significantly enriched GO terms included Cytosolic Ribosome (GO:0022626), Cytosolic Large Ribosomal Subunit (GO:0022625), Ribosome (GO:0005840), Cytosol (GO:0005829), and Cytoplasm (GO:0005737), which are associated with key components involved in protein synthesis and metabolic activities within the cytoplasm. In the molecular function (MF) category, significantly enriched GO terms were Structural Constituent of Ribosome (GO:0003735), Aspartic-Type Endopeptidase Activity (GO:0004190), Unfolded Protein Binding (GO:0051082), Antioxidant Activity (GO:0016209), and Peroxidase Activity (GO:0004601), indicating roles in protein structure, degradation, and antioxidative activities. In the biological process (BP) category, significantly enriched GO terms included Cytoplasmic Translation (GO:0002181), Protein Folding (GO:0006457), Purine Ribonucleoside Triphosphate Metabolic Process (GO:0009205), Drug Metabolic Process (GO:0017144), and Oxidative Phosphorylation (GO:0006119), which encompass processes related to protein synthesis, metabolic reorganization, and energy production. Additionally, this study analyzed the differentially expressed genes between regenerated mycelium and mycelium prior to cell wall removal (LDY1 vs. LDY3). GO annotation results indicated that 385 genes were annotated to biological processes (BP), 107 to cellular components (CC), and 128 to molecular functions (MF) (Figure 5c). In the cellular component (CC) category, significantly enriched terms included cytosolic ribosome (GO:0022626), cytosolic large ribosomal subunit (GO:0022625), ribosomal subunit (GO:0044391), ribosome (GO:0005840), and cytosol (GO:0005829), highlighting the ongoing protein synthesis and translation activities in the cytoplasm of the regenerated mycelium. In the molecular function (MF) category, enriched terms included structural constituent of ribosome (GO:0003735), unfolded protein binding (GO:0051082), proton channel activity (GO:0015252), proton-transporting ATP synthase activity, rotational mechanism (GO:0046933), and oxidoreductase activity (GO:0016491), reflecting the active roles of the regenerated mycelium in protein synthesis, energy metabolism, and antioxidant defense. In the biological process (BP) category, significantly enriched terms included cytoplasmic translation (GO:0002181), peptide biosynthetic process (GO:0043043), protein folding (GO:0006457), cell redox homeostasis (GO:0045454), and chaperone-mediated protein folding (GO:0061077), indicating the sustained requirements for protein synthesis, folding, and maintaining cellular redox balance in the regenerated mycelium. Through enrichment analysis of the shared GO terms during protoplast preparation and regeneration (LDY1 vs. LDY2 and LDY2 vs. LDY3), we found significant enrichment of ribosome-related GO terms within the Cellular Component category, including cytosolic ribosome, cytosolic large ribosomal subunit, ribosomal subunit, and cytosol. In the Molecular Function category, GO terms related to ribosomal structure and protein degradation, such as structural constituent of ribosome, structural molecule activity, aspartic-type endopeptidase activity, and aspartic-type peptidase activity, were significantly enriched. In the Biological Process category, the only shared enriched GO term was cytoplasmic translation. These common GO terms suggest that ribosomes are highly active during protoplast preparation and regeneration, with protein synthesis and quality control likely serving as core activities.
To further elucidate the biological pathways associated with these differentially expressed genes (DEGs), we performed KEGG enrichment analysis for LDY1 vs. LDY2, LDY2 vs. LDY3, and LDY1 vs. LDY3, selecting the top 30 enriched pathways, as shown in Figure 6. In the KEGG enrichment analysis of DEGs between LDY1 and LDY2, the most significantly enriched pathways included Ribosome (ko03010), Peroxisome (ko04146), Fatty acid degradation (ko00071), beta-Alanine metabolism (ko00410), Biosynthesis of unsaturated fatty acids (ko01040), Histidine metabolism (ko00340), Glycolysis/Gluconeogenesis (ko00010), Steroid biosynthesis (ko00100), and Tryptophan metabolism (ko00380). These enriched pathways indicate that during the removal of the fungal cell wall to form protoplasts, cells may adapt to structural changes and metabolic stress by regulating protein synthesis, energy metabolism, as well as lipid and amino acid metabolic activities. In the LDY2 vs. LDY3 stage, the primary enriched pathways included Ribosome (ko03010), Oxidative phosphorylation (ko00190), Proteasome (ko03050), Glutathione metabolism (ko00480), Peroxisome (ko04146), Various types of N-glycan biosynthesis (ko00513), Arginine and proline metabolism (ko00330), Glycosphingolipid biosynthesis - globo and isoglobo series (ko00603), and Protein processing in the endoplasmic reticulum (ko04141). The enrichment of these pathways suggests that, during the regeneration phase, protoplasts may maintain physiological homeostasis and ensure successful regeneration through coordinated regulation of protein synthesis, energy metabolism, protein degradation, and antioxidant mechanisms. Similarly, we performed KEGG enrichment analysis on DEGs for LDY1 vs. LDY3, which included pathways such as Ribosome (ko03010), Proteasome (ko03050), Glycolysis/Gluconeogenesis (ko00010), Steroid biosynthesis (ko00100), Pyruvate metabolism (ko00620), Oxidative phosphorylation (ko00190), Seleno compound metabolism (ko00450), Arginine and proline metabolism (ko00330), Glutathione metabolism (ko00480), and Endocytosis (ko04144). This indicates that regenerated mycelia maintain high metabolic activity in protein metabolism, energy generation, and antioxidant stress, ensuring functional recovery and maintenance of physiological homeostasis. Additionally, by analyzing KEGG pathways common to protoplast preparation and regeneration (LDY1 vs. LDY2 and LDY2 vs. LDY3), we identified that Ribosome (ko03010), Oxidative phosphorylation (ko00190), Peroxisome (ko04146), Arginine and proline metabolism (ko00330), Glycolysis/Gluconeogenesis (ko00010), Pyruvate metabolism (ko00620), Fatty acid degradation (ko00071), Fatty acid biosynthesis (ko00061), Taurine and hypotaurine metabolism (ko00430), Pantothenate and CoA biosynthesis (ko00770), beta-Alanine metabolism (ko00410), alpha-Linolenic acid metabolism (ko00592), Glyoxylate and dicarboxylate metabolism (ko00630), and Biosynthesis of unsaturated fatty acids (ko01040) were consistently enriched. This suggests that, whether during the removal of the fungal cell wall to form protoplasts or during the protoplast regeneration stage, protein synthesis, and antioxidant stress responses may play critical roles.”by “Gene Ontology (GO) includes three categories: molecular function (MF), cellular component (CC), and biological process (BP). GO analysis during protoplast preparation and regeneration is shown in Figure 5. The results showed that DEGs participate in multiple biologically significant processes. GO enrichment analysis for DEGs between protoplasts and the mycelial stage (LDY2 vs. LDY1) identified 51 CC, 92 MF, and 352 BP terms (Figure 5a). Enriched CC terms included Cytosolic Ribosome, Ribosome, Ribosomal Subunit, Cytosolic Large Ribosomal Subunit , Peroxisome , and Microbody, associated with protein synthesis and metabolic regulation. Enriched MF terms included Structural Constituent of Ribosome, Structural Molecule Activity, Aspartic-Type Endopeptidase Activity, RNA-Directed DNA Polymerase Activity, and Oxidoreductase Activity, linked to ribosomal structure and oxidative stress responses. BP terms included Cytoplasmic Translation, Translation, Peptide Biosynthetic Process, Cellular Response to Oxidative Stress, and Response to Oxidative Stress, highlighting involvement in protein synthesis and oxidative stress.For DEGs between the protoplast regeneration and protoplast stages (LDY2 vs. LDY3), 110 CC, 110 MF, and 407 BP terms were identified (Figure 5b). Enriched CC terms included Cytosolic Ribosome, Cytosolic Large Ribosomal Subunit, Ribosome, Cytosol, and Cytoplasm, essential for protein synthesis and metabolic activity. Enriched MF terms included Structural Constituent of Ribosome, Aspartic-Type Endopeptidase Activity, Unfolded Protein Binding, Antioxidant Activity, and Peroxidase Activity, reflecting roles in protein structure, degradation, and antioxidative responses. BP terms included Cytoplasmic Translation, Protein Folding, Purine Ribonucleoside Triphosphate Metabolic Process, Drug Metabolic Process, and Oxidative Phosphorylation, indicating protein synthesis, metabolic reorganization, and energy production.For DEGs between regenerated mycelium and mycelium before cell wall removal (LDY1 vs. LDY3), 107 CC, 128 MF, and 385 BP terms were identified (Figure 5c). Enriched CC terms included Cytosolic Ribosome, Cytosolic Large Ribosomal Subunit, Ribosomal Subunit, Ribosome, and Cytosol, highlighting protein synthesis in the cytoplasm. Enriched MF terms included Structural Constituent of Ribosome, Unfolded Protein Binding, Proton Channel Activity, Proton-Transporting ATP Synthase Activity, and Oxidoreductase Activity, reflecting roles in protein synthesis, energy metabolism, and antioxidant defense BP terms included Cytoplasmic Translation, Peptide Biosynthetic Process, Protein Folding, Cell Redox Homeostasis, and Chaperone-Mediated Protein Folding, highlighting protein synthesis, folding, and redox balance.Shared GO terms during protoplast preparation and regeneration (LDY1 vs. LDY2 and LDY2 vs. LDY3) showed significant enrichment of ribosome-related CC terms, including Cytosolic Ribosome, Cytosolic Large Ribosomal Subunit, Ribosomal Subunit, and Cytosol. Enriched MF terms included Structural Constituent of Ribosome, Structural Molecule Activity, Aspartic-Type Endopeptidase Activity, and Aspartic-Type Peptidase Activity. The only shared BP term was Cytoplasmic Translation, These common GO terms suggest that ribosomes are highly active during protoplast preparation and regeneration, with protein synthesis and quality control likely serving as core activities.
To further elucidate the biological pathways associated with these DEGs, we conducted KEGG enrichment analysis for comparisons between LDY1 vs. LDY2, LDY2 vs. LDY3, and LDY1 vs. LDY3, selecting the top 30 enriched pathways (Figure 6). In the KEGG enrichment analysis of DEGs between LDY1 and LDY2(Figure 6a), the most significantly enriched pathways included Ribosome, Peroxisome, Fatty Acid Degradation, Beta-Alanine Metabolism, Biosynthesis of Unsaturated Fatty Acids, Histidine Metabolism, Glycolysis/Gluconeogenesis, Steroid Biosynthesis, and Tryptophan Metabolism. These enriched pathways indicate that during the removal of the fungal cell wall to form protoplasts, cells may adapt to structural changes and metabolic stress by regulating protein synthesis, energy metabolism, as well as lipid and amino acid metabolic activities.In the LDY2 vs. LDY3 stage(Figure 6b), the primary enriched pathways included Ribosome , Oxidative Phosphorylation, Proteasome, Glutathione Metabolism, Peroxisome , Various Types of N-Glycan Biosynthesis, Arginine and Proline Metabolism, Glycosphingolipid Biosynthesis - Globo and Isoglobo Series, and Protein Processing in the Endoplasmic Reticulum. The enrichment of these pathways suggests that, during the regeneration phase, protoplasts may maintain physiological homeostasis and ensure successful regeneration through coordinated regulation of protein synthesis, energy metabolism, protein degradation, and antioxidant mechanisms.Similarly, KEGG enrichment analysis of DEGs between LDY1 and LDY3 (Figure 6c) identified pathways including Ribosome, Proteasome, Glycolysis/Gluconeogenesis, Steroid Biosynthesis, Pyruvate Metabolism, Oxidative Phosphorylation, Seleno Compound Metabolism, Arginine and Proline Metabolism, Glutathione Metabolism, and Endocytosis. This indicates that regenerated mycelia maintain high metabolic activity related to protein metabolism, energy generation, and antioxidant stress, ensuring functional recovery and physiological homeostasis.Additionally, by analyzing KEGG pathways shared between protoplast preparation and regeneration (LDY1 vs. LDY2 and LDY2 vs. LDY3), we found consistent enrichment of pathways such as Ribosome, Oxidative Phosphorylation, Peroxisome, Arginine and Proline Metabolism, Glycolysis/Gluconeogenesis, Pyruvate Metabolism, Fatty Acid Degradation, Fatty Acid Biosynthesis, Taurine and Hypotaurine Metabolism, Pantothenate and CoA Biosynthesis, Beta-Alanine Metabolism, Alpha-Linolenic Acid Metabolism, Glyoxylate and Dicarboxylate Metabolism, and Biosynthesis of Unsaturated Fatty Acids. This suggests that, whether during fungal cell wall removal to form protoplasts or during protoplast regeneration, protein synthesis and antioxidant stress responses may play critical roles.”
Comments 2: It is necessary to write the Conclusion section more clearly and briefly.
Response 2: Thank you for your suggestion. We have revised the conclusion section by removing unnecessary content and simplifying the statement to make it clearer and more concise, ensuring that the key findings are effectively highlighted.
Page 18-19, paragraph 5,Line 650-667:Change “In this study, we optimized the conditions for the preparation and regeneration of the protoplasts of L decastes, and analysed the changes in gene expression at different stages of preparation in combination with transcriptomics. Our one-way experiments clarified the factors that significantly impacted the preparation of protoplasts of strain HMJU 6921, including the type of osmotic stabiliser, enzyme concentration, Digestion Temperature and digestion time, and mycelial age. Orthogonal experiments showed that the optimal preparation conditions were 10 days of mycelial age, 2.25 h of digestion time, 2% of enzyme concentration, and 28°C of digestion Temperature, and the amount of protoplast preparation could reach 5.475 × 10^6 cells/mL. The regeneration rate of the regeneration medium Z5 with the addition of L. decastes mycelial extract could reach up to 2.86%. Transcriptomics analysis showed that several key biological pathways and genes involving cell wall synthesis, stress response, and metabolic regulation were significantly changed during the preparation and regeneration of protoplasts. This study provides a theoretical basis and technical support for the efficient preparation and regeneration of L. decastes protoplasts and offers important gene expression information for further functional studies and application development, such as genetic modification and breeding of L. decastes. These findings open up exciting possibilities for further research and application in the field, sparking curiosity and eagerness for what the future holds.” by “In this study, we optimized the conditions for protoplast preparation and regeneration in L. decastes and analyzed gene expression changes at different preparation stages using transcriptomics. One-way experiments identified key factors affecting protoplast preparation in strain HMJU 6921, including the type of osmotic stabilizer, enzyme concentration, digestion temperature, digestion time, and mycelial age. Orthogonal experiments determined the optimal conditions: 10-day-old mycelia, 2.25 hours of digestion, 2% enzyme concentration, and 28°C digestion temperature, yielding 5.475 × 10^6 cells/mL of protoplasts. The regeneration rate on medium Z5 supplemented with L. decastes mycelial extract reached 2.86%. Transcriptomic analysis revealed significant changes in key biological pathways and genes related to cell wall synthesis, stress responses, and metabolic regulation during protoplast preparation and regeneration. This study provides theoretical and technical support for efficient L. decastes protoplast preparation and regeneration, offering valuable insights for further research, including genetic modification and breeding, and advancing the application potential of L. decastes.”
Comments 3: It is necessary to correct the signature in Fig. 8.
Response 3: Thank you very much for your pointing out. We are very sorry that the signature of Figure 8 was wrong due to our negligence. We have corrected the signature of Figure 8.
Page 14, paragraph 3.9,Line 463: Add “downregulation.”
Comments 4: It is necessary to provide a separate subsection on the statistical method in the Materials and Methods section
Response 4: Agree,Thank you for your suggestion. We have added a separate subsection on Statistical Methods in the Materials and methods section to clearly and in detail describe the statistical analysis methods used in the study. This revision ensures that the statistical analysis is clearly outlined and easy to reference.
Page 6, paragraph 2.11,Line 191-194:Add “2.11 Statistical Analysis
Orthogonal experimental design and range analysis of the results were performed using IBM SPSS Statistics 26. Single-factor analysis of variance (ANOVA) was conducted with GraphPad Prism 10, which was also utilized for figure generation.”
Comments 5: It is necessary to illustrate the content of ribosomes in the cell during the preparation of protoplasts using the electron microscopic method.
Response 5: Thank you for your valuable suggestions. We will use electron microscopy to show the changes of ribosomes and other related organelles during protoplast preparation and regeneration in another study "Study on the Regeneration Mechanism of Protoplasts in lyophyllum decastes". If it is included in this study, it may affect the integrity of the other study. I hope to get your guidance again in the other article.
Comments 6: It is desirable to provide the cellular response to oxidative stress by ROS production.
Response 6: Thank you for your valuable suggestions. Our study mainly focused on changes in transcriptional levels, and only speculated in the discussion section that cells may respond to oxidative stress to adapt to changes in cell morphology during protoplast preparation and regeneration, without directly measuring the production of ROS. We hope that future studies will improve the integrity of this part of the experiment to better explain the changes in these pathways at the transcriptional level. We believe this will be a huge project.
Comments 7: It is desirable to describe glutathione metabolism in more detail.
Response 7: Thank you for your valuable suggestions. We revised the manuscript to provide a more detailed description of glutathione metabolism, emphasizing its role in maintaining redox balance, neutralizing reactive oxygen species (ROS), and ensuring cell protection during regeneration.
Page 17, paragraph 4,Line 597-600:Add “In particular, 23 of the 30 DEGs annotated to the glutathione metabolic pathway were upregulated. Glutathione, a tripeptide thiol present in almost all cells, is an important metabolite in eukaryotes and plays an important role in metabolism, transport, and protection of cells from oxidative damage [66-68].”
Comments 8: It is necessary to bring the list of references in line with the requirements of the journal.
Response 8: Thank you for pointing this out. We have revised the reference list to ensure that it complies with the journal’s formatting and citation requirements.
Additional clarifications
We have updated the latest production of L. decastes.
Page 1, paragraph 1,Line 39-40:Change “The production of L. decastes in China reached 960 tons in 2014 and experienced rapid growth after 2015, reaching 21,805 tons in 2021 [8,9].” By “The production of L. decastes in China reached 960 tons in 2014 and experienced rapid growth after 2015 [8,9]. According to the data of the public service platform of the China Edible Fungi Association (https://bigdata.cefa.org.cn/output.html), the production reached 107,601 tons in 2022.”
Reviewer 2 Report
The manuscript presents significant relevance to the scientific community. I have conducted a thorough verification of both the online databases and the document itself. Only minor adjustments and clarifications are required to enhance the quality of the document. Also, try to reduce the similarity score according to the ithenticate software to 15% or less.
Please note the following points for clarity and completeness:
1. In line 85, it is essential to provide a detailed definition of osmotic solutions, including their concentrations, technical grade (whether reagent or molecular biology grade), brand, and batch number.
2. Additional technical information regarding the filters utilized in the process should be included to enhance understanding.
3. In line 87, please ensure that enzymatic activity is expressed in the appropriate measurement units, as percentages are not suitable for this purpose.
4. A table summarizing the quality and concentration of the obtained RNA would be beneficial.
5. If feasible, please include gel images that demonstrate the integrity of the RNA for further verification.
Author Response
Thank you for your comments concerning our manuscript entitled “Optimization of Protoplast Preparation Conditions in Lyophyllum decastes and Transcriptomic Analysis Throughout the Process”. Those comments were all valuable and very helpful in revising and improving our manuscript. We have studied the comments carefully and have made corrections which we hope meet with your approval. Revised portions are indicated in red. The main corrections in the paper and the response to the reviewer’s comments are as follows:
Comments 1: In line 85, it is essential to provide a detailed definition of osmotic solutions, including their concentrations, technical grade (whether reagent or molecular biology grade), brand, and batch number.
Response 1: Thank you for your suggestion. We have updated the manuscript to include a detailed definition of the osmotic solutions used, specifying their concentrations, technical grade (reagent or molecular biology grade), brand, and batch number . This ensures clarity and reproducibility of the experimental procedures.
Page 2, paragraph 2.2,Line 85:Add “(The osmotic stabilizer is prepared by dissolving the corresponding inorganic salt in sterile water at a concentration of 0.6 mol·L⁻¹, sterilized at 121℃ and0.1 MPa for 15 minutes and then used. Aladdin™ S112236 Sucrose 99.5%, Aladdin™ D657382 Mannitol 97%, Aladdin™ C111538 Sodium chloride 99.5%, Aladdin™ P301833 Potassium chloride 99%, Aladdin™ M119381 Magnesium sulfate anhydrous 99%)”
Comments 2: Additional technical information regarding the filters utilized in the process should be included to enhance understanding.
Response 2: Thank you for your suggestion. We have revised the manuscript to include detailed information about the filtration process.
Page 2, paragraph 2.2,Line 85: Add “(using sterile double gauze to filter)”
Page 2, paragraph 2.2,Line 91: Change “The extract was filtered using sterile absorbent cotton ” by “Plug the top of a 2 mL glass syringe with approximately 1 cm of sterile absorbent cotton. After sterilization, use it to filter the extract.”
Comments 3: In line 87, please ensure that enzymatic activity is expressed in the appropriate measurement units, as percentages are not suitable for this purpose.
Response 3: Thank you for your valuable suggestions. Lywallzyme is produced and provided by Guangdong Microbial Culture Collection Center. The product is light brown powder. According to the product instructions, the corresponding osmotic pressure stabilizer is used to prepare a lysing enzyme solution with a mass concentration of 1.25-2.25% (W/V) for enzymatic hydrolysis of fungal cell walls. We have added the Enzyme Concentration percentage in the manuscript to indicate the mass concentration.
Page 3, paragraph 2.3,Line 100: Add “(Mass concentration m/v)”
Comments 4: A table summarizing the quality and concentration of the obtained RNA would be beneficial.
Response 4: Thank you for your suggestion. We have added a supplementary Table S1 summarizing the quality and concentration of the obtained RNA, including key indicators such as Nanodrop Concentration, Total Amount , OD260/280, to provide a clearer overview of the quality of the transcriptome RNA.
Page 9, paragraph 3.4,Line 279: Add “The RNA quality test results are shown in Table S1, and the gel image is shown in Figure S1. ”
Page 19, paragraph Supplementary Materials,Line 668: Add “Supplementary Materials: The following supporting information can be downloaded at: www.mdpi.com/xxx/s1, Figure S1: RNA agarose gel electrophoresis results; Table S1: RNA Quality Test Results.”
Comments 5:If feasible, please include gel images that demonstrate the integrity of the RNA for further verification.
Response 5: Thank you for your suggestion. We have included gel images figure S1 in the revised manuscript to demonstrate the integrity of the RNA.
Page 9, paragraph 3.4,Line 279: Add “The RNA quality test results are shown in Table S1, and the gel image is shown in Figure S1. ”
Page 19, paragraph Supplementary Materials,Line 668: “Supplementary Materials: The following supporting information can be downloaded at: www.mdpi.com/xxx/s1, Figure S1: RNA agarose gel electrophoresis results; Table S1: RNA Quality Test Results.”
Response to Comments on the Quality of images
Point 1: The quality of the images needs to be improved.
Response 1: Thank you for your suggestion. We are very sorry, we have completed the content of Figure 7 in the manuscript.
Page 13, paragraph 3.7,Line 418:
Change
by
Round 2
Reviewer 2 Report
Dear Authors,
Thank you for your attention to my comments and observations. The changes you have made significantly improve the quality and scientific rigor of the manuscript. Regarding to contained, the manuscript is now suitable for publication. However, the similarity percentage must be reduced to below 15%.
I detected that some sections of the protocols have not been changed from their original sources. If this is the case, I recommend a change in the manuscript; that the protocol was carried out without modifications according to [original source], and that the complete protocol should be included in the supplementary materials in a reader-friendly format.
Author Response
Thank you for your valuable comments on our manuscript entitled "Optimization of Protoplast Preparation Conditions in Lyophyllum decastes and Transcriptomic Analysis Throughout the Process". We have carefully studied these comments and made further corrections, hoping to gain your approval. The revised parts are marked in red. The main corrections in the paper and the responses to the reviewers' comments are as follows:
Comments 1: Thank you for your attention to my comments and observations. The changes you have made significantly improve the quality and scientific rigor of the manuscript. Regarding to contained, the manuscript is now suitable for publication. However, the similarity percentage must be reduced to below 15%.
Response 1: Thank you for your valuable suggestions and recognition of the manuscript improvement. We have further revised the manuscript to ensure that the similarity is reduced to less than 15%. We sincerely appreciate your guidance throughout the review process.
Page 1, paragraph 1,Line 30-31 :Change “Lyophyllum decastes, also known as the fried chicken mushroom, is a rare edible fungus that's a nutritional powerhouse.” By “Lyophyllum decastes, commonly referred to as the fried chicken mushroom, is a rare edible fungus renowned for its exceptional nutritional value.”
Page 2, paragraph 2.1,Line 75-83 :Change “The fruiting bodies of L decastes used in this study were collected from the Qipan Mountain National Scenic Tourist Area in Shenyang, Liaoning Province. The strain "HMJU 6921" was obtained through tissue isolation and preserved at the Fungal Resource Development and Utilization Laboratory of Jilin Agricultural Science and Technology College following identification. It was inoculated onto potato dextrose agar (PDA) medium and incubated at a constant temperature of 25°C in the dark for 10 days. Using a sterile scalpel, the cultured strain was cut into small pieces (approximately 6 mm × 6 mm), inoculated into 100 mL of potato dextrose broth, and incubated at 120 rpm and 25°C in the dark. ”By “The fruiting bodies of L. decastes utilized in this research were sourced from the Qipan Mountain National Scenic Tourist Area, located in Shenyang, Liaoning Province. The strain "HMJU 6921" was isolated from tissue samples, identified, and stored at the Fungal Resource Development and Utilization Laboratory of Jilin Agricultural Science and Technology College. It was cultivated on potato dextrose agar (PDA) medium and maintained at a stable temperature of 25°C under dark conditions for 10 days. After in-cubation, small sections of the cultured strain, approximately 6 mm × 6 mm in size, were excised using a sterile scalpel, transferred into 100 mL of potato dextrose broth, and incubated in darkness at 25°C while shaken at 120 rpm.”
Page 3, paragraph 2.5,Line 118-123 :Change “The protoplast stock solution was diluted to an appropriate concentration using 0.6 mol·L⁻¹ osmotic stabilizer, and 100 μL of the diluted solution was evenly spread onto different regeneration media (Table 3), with PDA medium used as the control group. Both the experimental and control groups were incubated at 25°C in the dark. Colony regeneration was observed daily, and the regeneration rate was calculated. Each exper-iment was repeated at least three times. The regeneration rate was calculated as follows:” by “The protoplast stock solution was adjusted to a suitable concentration by diluting it with a 0.6 mol·L⁻¹ osmotic stabilizer. Subsequently, 100 μL of this diluted solution was evenly distributed across various regeneration media (refer to Table 3), while PDA medium served as the control. Both the experimental and control setups were maintained at 25°C in darkness. Daily observations were made to monitor colony regeneration, and the regeneration rate was determined. Each experiment was performed a minimum of three times, and the regeneration rate was calculated using the following formula:”
Page 11, paragraph 3.6,Line 388-389 :Change “Figure 5 Gene Ontology (GO) Enrichment Analysis of Differentially Expressed Genes During Pro-toplast Preparation and Regeneration. (a) LDY1 VS LDY2. (b) LDY2 VS LDY3. (c) LDY1 VS LDY3.” By “Figure 5 GO Enrichment Analysis of DEGs During Pro-toplast Preparation and Regeneration. (a) LDY1 VS LDY2. (b) LDY2 VS LDY3. (c) LDY1 VS LDY3.”
Page 11, paragraph 3.6,Line 391-393 :Change “Figure 6 Kyoto Encyclopedia of Genes and Genomes (KEGG) Pathway Enrichment Analysis of Differentially Expressed Genes (DEGs) During Protoplast Preparation and Regeneration. (a) LDY1 VS LDY2. (b) LDY2 VS LDY3. (c) LDY1 VS LDY3.” By “Figure 6 KEGG Pathway Enrichment Analysis of DEGs During Protoplast Preparation and Regeneration. (a) LDY1 VS LDY2. (b) LDY2 VS LDY3. (c) LDY1 VS LDY3.”
Page 12, paragraph 3.7,Line 395-398 :Change “To validate the transcriptome data, qRT-PCR was used to analyze the expression patterns of genes related to protoplast preparation and regeneration processes. The re-sults were consistent with those obtained from RNA-Seq, confirming the reliability of our RNA-Seq data through the consistency between the two methods (Figure 7a).” by “To confirm the transcriptome data, qRT-PCR was performed to examine the ex-pression patterns of genes associated with protoplast preparation and regeneration. The qRT-PCR results aligned with the RNA-Seq findings, demonstrating the reliability of our RNA-Seq data through the concordance of the two methods (Figure 7a).”
Comments 2: I detected that some sections of the protocols have not been changed from their original sources. If this is the case, I recommend a change in the manuscript; that the protocol was carried out without modifications according to [original source], and that the complete protocol should be included in the supplementary materials in a reader-friendly format.
Response 2: Thank you for your valuable suggestions. We have revised the manuscript to clearly state that the protocol was performed without modification according to [Wang, T.; Li, X.; Zhang, C.; Xu, J. Transcriptome analysis of Ganoderma lingzhi (Agaricomycetes) response to Trichoderma hengshanicum infection. Front Microbiol 2023, 14, 1131599, doi:10.3389/fmicb.2023.1131599.]. In addition, we have included the complete protocol (Scheme S1) and primer sequences for qRT-PCR (Table S2) in the supplementary material.
Page 5-6, paragraph 2.7-2.10,Line 145-197 :Change “2.7 RNA Extraction and Sequencing
Total RNA was isolated using the Trizol Reagent (Invitrogen Life Technologies), after which the concentration, quality, and integrity were determined using a NanoDrop spectrophotometer (Thermo Scientific). Sequencing libraries were generated using the TruSeq RNA Sample Preparation Kit (Illumina, San Diego, CA, USA). mRNA was purified from total RNA using poly-T oligo-attached magnetic beads. Then the enriched mRNA was fragmented into short fragments using a fragmentation buffer and reverse transcribed into cDNA with random primers. The library fragments were purified using the AMPure XP system (Beckman Coulter, Beverly, CA, USA). DNA fragments with ligated adaptor molecules on both ends were selectively enriched using Illumina PCR Primer Cocktail. Products were purified (AMPure XP system) and quan-tified using the Agilent high-sensitivity DNA assay on a Bioanalyzer 2100 system (Ag-ilent). The sequencing library was then sequenced on a NovaSeq 6000 platform (Illu-mina) by Shanghai Personal Biotechnology Co., Ltd.
2.8 De novo Transcriptome Assembly and Gene Annotation
Samples are sequenced on the platform to get image files, which are transformed by the software of the sequencing platform, and the original data in FASTQ format (Raw Data) is generated. Sequencing data contains a number of connectors and low-quality reads, so we use fastp (v0.22.0) software to filter the sequencing data to get high-quality sequences (Clean Reads) for further analysis. For the transcriptome se-quencing project without a reference genome, we use Trinity (v2.15.1) [23] software to assemble Clean Reads for transcripts for later analysis. After the completion of assem-bly, transcript sequence files in FASTA format can be obtained. The longest transcript of each gene was extracted as the representative sequence of the gene, called Unigene. We have annotated gene functions for Unigenes. The databases used in gene function annotation include NR (NCBI non-redundant protein sequences), GO (Gene Ontology), KEGG (Kyoto Encyclopedia of Genes and Genomes), eggNOG (evolutionary genealogy of genes: Non-supervised Orthologous Groups), Swiss-Prot, and Pfam.
2.9 Differential Gene Expression Analysis and Functional Enrichment
Using RSEM (v2.15) statistics, we compared the Read Count values for each gene to represent its original expression level, and FPKM was used to normalize gene ex-pression levels. Three pairwise comparisons were conducted using RNA-seq data, spe-cifically between LDY1 vs. LDY2, LDY2 vs. LDY3, and LDY1 vs. LDY3. Next, we used DESeq (v1.38.3) to analyze the differentially expressed genes (DEGs), with screening criteria of | log2FoldChange |> 1 and a significant P-value < 0.05. All DEGs were mapped to terms in the Gene Ontology database, and the number of enriched DEGs was calcu-lated for each term. GO enrichment analysis was performed using topGO (v2.50.0), with P-values calculated through the hypergeometric distribution method, where sig-nificant enrichment was defined by a P-value < 0.05. This enabled the identification of GO terms with significantly enriched DEGs, thereby determining the primary biologi-cal functions of these genes. KEGG pathway enrichment analysis of DEGs was per-formed using ClusterProfiler (v4.6.0), focusing on significantly enriched pathways (P-value < 0.05).
2.10 Quantitative real-time polymerase chain reaction (qRT-PCR) verification
The RNA-seq results were validated by selecting 11 DEGs to assess the consistency of their expression patterns. Total RNA was isolated using the Trizol Reagent (Invitro-gen Life Technologies) and reverse-transcribed into cDNA with the PrimeScriptTM 1st Strand cDNA Synthesis Kit. Glyceraldehyde 3-phosphate dehydrogenase (GAPDH) was used as the housekeeping gene[24]. The reaction system included 10 μl of 2× SYBR real-time PCR premixture, 0.4 μl each of forward and reverse primers, and 1 μl of cDNA, with RNase-free dH2O added to a total volume of 20 μl. The qRT-PCR program was set as follows: 95°C for 5 minutes, followed by 40 cycles of 95°C for 15 seconds and 60°C for 30 seconds, with each sample tested in triplicate. Relative gene expression lev-els were calculated using the 2^-ΔΔCt method[25].” by "
“2.7 Transcriptome sequencing, assembly, DEG functional annotation and qRT-PCR validation
All experimental protocols, including RNA extraction, quality assessment, library preparation, de novo transcriptome assembly, DEG analysis, functional enrichment and qRT-PCR verification, were performed according to the original protocol of Wang Tiantian [23]without modification. The complete protocol is included in the supplemen-tary material (Scheme S1),and the qRT-PCR primer sequences have been placed in Table S2. References [24,25]are cited in the Supplementary Materials”
Page 6, paragraph 2.11,Line198 :Change “2.11 Statistical Analysis” by “2.8 Statistical Analysis”
Page 19, paragraph Supplementary Materials,Line 651: Add “Scheme S1: Original Protocol for Transcriptome sequencing, assembly, DEG functional annotation and qRT-PCR validation; Table S2: qRT-PCR primer sequences.”